# Complete Structure Guided Point Cloud Completion via Cluster- and Instance-Level Contrastive Learning

**Yang Chen**[1], **Yirun Zhou**[1], **Weizhong Zhang**[2,3], **Cheng Jin**[1,3*]

[1]College of Computer Science and Artificial Intelligence
[2]School of Data Science, Fudan University
[3]Shanghai Key Laboratory of Intelligent Information Processing
{chen_yang23,yrzhou22}@m.fudan.edu.cn, {weizhongzhang,jc}@fudan.edu.cn

## Abstract

Point cloud completion, aiming to reconstruct missing part from incomplete point clouds, is a pivotal task in 3D computer vision. Traditional supervised approaches often necessitate complete point clouds for training supervision, which are not readily accessible in real-world applications. Recent studies have attempted to mitigate this dependency by employing self-supervise mechanisms. However, these approaches frequently yield suboptimal results due to the absence of complete structure in the point cloud data during training. To address these issues, in this paper, we propose an effective framework to complete the point cloud under the guidance of self learned complete structure. A key contribution of our work is the development of a novel self-supervised complete structure reconstruction module, which can learn the complete structure explicitly from incomplete point clouds and thus eliminate the reliance on training data from complete point clouds. Additionally, we introduce a contrastive learning approach at both the cluster- and instance-level to extract shape features guided by the complete structure and to capture style features, respectively. This dual-level learning design ensures that the generated point clouds are both shape-completed and detail-preserving. Extensive experiments on both synthetic and real-world datasets demonstrate that our approach significantly outperforms state-of-the-art self-supervised methods.

## 1 Introduction

The advancement and widespread adoption of 3D sensors, particularly LiDAR, have led to the emergence of point clouds as a dominant representation of 3D shapes across various domains [6, 10] primarily owning to their ease of acquisition and comprehensive geometric features. However, in real-world scenarios, raw point clouds often suffer from incompleteness caused by factors such as self-occlusion and lighting conditions, which can hinder the performance of downstream tasks, including object detection [22] and segmentation [4]. Consequently, point cloud completion has been proposed as an effective solution to infer complete point clouds from incomplete inputs.

Supervised learning is the paradigm adopted by most existing methods [38, 41, 27]. These approaches typically utilize paired complete and incomplete point cloud data to train neural networks, learning a one-to-one mapping from incomplete point clouds to their corresponding complete counterparts. Impressive results have been reported in recent studies. However, since high-quality complete point clouds are often difficult to obtain through real-world scanning, paired data for training is frequently sourced from virtual datasets, such as ShapeNet [2]. As a result, supervised methods contend with the domain gap when applied to real-world data.

---

*Corresponding author

39th Conference on Neural Information Processing Systems (NeurIPS 2025).

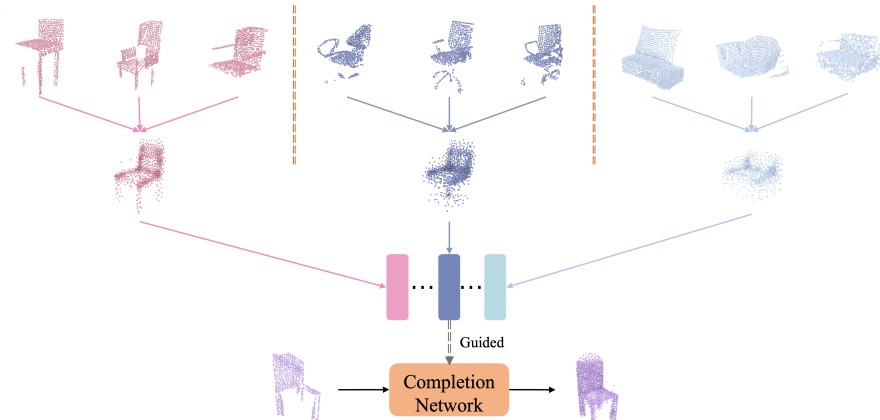

Figure 1: Illustration of ideas. Our method clusters incomplete point clouds using contrastive learning. By aggregating point clouds in the clusters, we obtain the complete structure in a self-supervised manner, which are used to guide the completion of the point cloud.

To address the above issue, unpaired point cloud completion methods [5, 39, 40] have been proposed as a potential solution. These methods eliminate the need for paired data, yet they still rely on the complete point clouds from virtual datasets, and thus can not fundamentally resolve the problem. Subsequently, weakly supervised methods [13, 26] were introduced, aiming to leverage the consistency of incomplete point clouds from multiple views. However, their effectiveness is highly dependent on registration accuracy, and collecting point clouds from different viewpoints poses significant challenges in real-world scenarios. Recently, several studies have sought to address this issue using a self-supervised paradigm [17, 7], with P2C approach [7] achieving breakthrough results. Nevertheless, due to the absence of guidance from complete structures, the completed point clouds generated by these methods remain suboptimal.

In this paper, we propose a novel self-supervised point cloud completion framework, termed **C**omplete **S**tructure **G**uided **P**oint **C**loud **C**ompletion (CSG-PCC). Our method follows an encoder-to-decoder pipeline, while the core innovation lies in adding an effective Complete Structure Reconstruction Module between the encoder and decoder. This module consists of two key components: 1) a feature disentanglement module and 2) a prototype projection module. For feature disentanglement module, we disentangle the point cloud features into two distinct dimensions, i.e., shape and style. The shape feature captures the global structure while the style feature extracts the local details. In the prototype projection module, we aggregate complete structures from the disentangled shape features to form shape prototypes, which are then used to guide point cloud completion by projected the shape feature to prototypes.

To ensure effective feature disentanglement critical for structure aggregation, we propose a Feature Permutation Consistency Constraint (FPCC). Within FPCC, we randomly swap point clouds' skeleton points obtained through farthest point sampling (FPS) and recompute their $k$-nearest neighbors to generate recombined point clouds. The FPCC enforces consistency between original and perturbed point clouds in both shape and style feature spaces, effectively decoupling the shape and style features. Building upon this disentanglement module, we construct a two-branch network architecture and design a dual-level contrastive learning method. Specifically, we employ cluster-level contrastive learning to cluster the shape features of the incomplete point clouds. By aggregating similar shapes in each cluster, we learn a corresponding complete structure, as shown in Figure 1. Additionally, we use instance-level contrastive learning on the results of the two branches to ensure that the learned style features focus on instance-specific details.

We conducted experiments with CSG-PCC on both real-world and synthetic datasets to validate the effectiveness of our design. Experimental results demonstrate that the dual-level contrastive learning design enables our method to generate point clouds that are both shape-completed and detail-preserving, achieving state-of-the-art performance in the self-supervised point cloud completion domain. Our main contributions can be summarized as:

1. We propose a novel self-supervised point cloud completion method, CSG-PCC, which can explicitly extract complete structures to guide the completion process.

2. We propose a dual-level contrastive learning framework to enable efficient training of our self-supervised point cloud completion network.

3. Extensive experiments on both synthetic and real-world datasets demonstrate that our method can significantly outperform state-of-the-art methods.

## 2 Related Work

### 2.1 Supervised and Unpaired Point Cloud Completion

Traditional point cloud completion methods can be broadly classified into those that leverage geometric priors [30, 19] and those that utilize template matching techniques [23, 28]. With the rise of deep learning, several methods have drawn inspiration from 2D image inpainting techniques and applied them to 3D point cloud completion. These methods [9, 14] typically use 3D convolutional networks to process voxelized point clouds. However, as the resolution of voxel grids increases, the computational cost grows significantly. The introduction of PointNet [24], which directly processes raw point clouds without the need for voxelization or other transformations, marked a significant shift in the field. Then deep learning methods [38, 29, 34, 27] have achieved significant success in the field of point cloud completion. Despite these advancements, the dependence of supervised point cloud completion methods on fully-complete point clouds remains a significant limitation, hindering their application in real-world scenarios. To reduce the dependency on paired data, unpaired methods [5, 31, 39, 11] have been proposed. However, these methods still rely on complete point cloud data, limiting their applicability in real-world scenarios.

### 2.2 Weakly-Supervised and Self-Supervised point cloud Completion

In contrast to previous methods, approaches [13, 21] have proposed weakly supervised paradigms to address the data dependency on complete point clouds. [13] utilizes incomplete point clouds from multiple viewpoints to predict the complete point cloud, using geometric consistency across different views as a constraint. However, these methods require point clouds to be captured from multiple viewpoints, which is not always feasible in real-world scenarios. Later, ACL-SPC [17], as a self-supervised method, introduced the concept of an adaptive cycle system. Inspired by [15], P2C [7] apply a mask-reconstruction self-supervised paradigm to the point cloud completion task. The self-supervised methods mitigate the dependency of complete point clouds. However, they didn't fully exploit the structural information corresponding to the complete structure, which limited the performance of completion results. Though [32] adopts a self-supervised approach, it requires additional depth images. [20] utilizes an alternative 3D representation, but our research focus on point cloud.

### 2.3 Contrastive Learning

Contrastive learning has become a powerful paradigm in unsupervised and self-supervised learning tasks, particularly in the domain of representation learning. Early work in contrastive learning [3, 16] focused on learning effective representations by contrasting positive and negative pairs of samples in a latent space. Later, contrastive learning methods [1, 12] further advanced the field by eliminating the need for negative samples. Contrastive learning has also been explored for point cloud [35, 36]. In this work, we propose to introduce contrastive learning into the self-supervised point cloud completion task.

## 3 Method

For point cloud completion, achieving a complete structure is critical for helping the missing region prediction. However, in self-supervised point cloud completion, the lack of complete point clouds makes it challenging to obtain the necessary structural information. In this paper, we propose a novel self-supervised point cloud completion framework, CSG-PCC, which leverages self-learned complete structures to guide the completion process. The overall pipeline of our network is illustrated

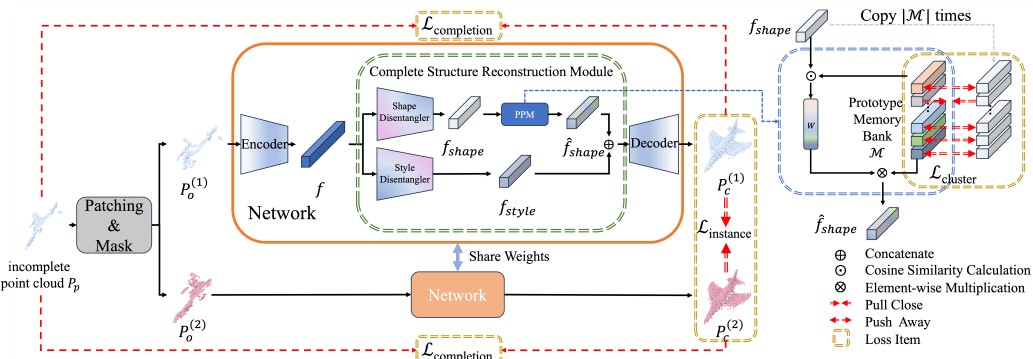

Figure 2: The Pipeline of CSG-PCC. Given a partial point cloud $P_p$, we divide it into patches, followed by random masking to create two incomplete point clouds, $P_o^{(1)}$ and $P_o^{(2)}$. Then point clouds are processed through the encoder to extract features, and then fed into the Complete Structure Reconstruction Module. The CSRM are composed of two core components: a) Feature Disentanglement Module: maps encoder outputs into shape features $f_{shape}$ and style features $f_{style}$ via two disentanglers. b) Prototype Projection Module: Refines $f_{shape}$ via learnable prototype memory bank $\mathcal{M}$, producing structure-enhanced features $\hat{f}_{shape}$. Then we concatenates $\hat{f}_{shape}$ and $f_{style}$ as decoder input to generate the completed point clouds. Dual-level contrastive learning are used to ensure structural completeness and detail preservation.

in Figure 2. In the following, we will present our method by introducing the framework especially the complete structure reconstruction module, cluster- and instance-level learning schemes, and the overall optimization in order.

## 3.1 Framework

As shown in the Figure 2, our contrastive learning-based framework maintains two identical branches during training, with each branch following the encoder-to-decoder pipeline. Through the Patching & Mask, we obtain distinct inputs for the two branches, with specific implementation details as follows:

**Patching & Mask.** Let $P_p \in \mathbb{R}^{N_p \times 3}$ be an input partial point cloud. We first divide the point cloud into $K$ patches using farthest point sampling (FPS) [25], where $k$-nearest neighbors are gathered around each sampled center. Then We randomly select $K_o$ observable patches from the total $K$ patches, then discard the remaining $K - K_o$ patches through masking to generate the first incomplete observable point cloud $P_o^{(1)} \in \mathbb{R}^{N_o \times 3}$ for Branch 1. Subsequently, we repeat the masking procedure with a different set of $K - K_o$ patches removed, thereby producing the second incomplete observable point cloud $P_o^{(2)} \in \mathbb{R}^{N_o \times 3}$ for Branch 2. Both processed point clouds serve as parallel inputs for network training.

Taking Branch 1 as an exemplar, the encoder first extracts a point cloud feature $f \in \mathbb{R}^D$ from input $P_o^{(1)}$, formally expressed as $f = E(P_o^{(1)})$. Our key innovation lies in the proposed **Complete Structure Reconstruction Module (CSRM)** strategically positioned between the encoder and decoder. The CSRM comprises two coordinated submodules:

1) **Feature Disentanglement Module** : Composed of two parallel disentanglers, this module decouples $f$ into shape feature $f_{shape} \in \mathbb{R}^{D/2}$ and $f_{style} \in \mathbb{R}^{D/2}$.

2) **Prototype Projection Module (PPM)**: Enhances structural completeness through geometric priors: $\hat{f}_{shape} = \mathcal{PPM}(f_{shape})$.

The refined shape feature $\hat{f}_{shape}$ is then concatenated with $f_{style}$ and fed to the decoder $D$ to generate the completed point cloud $P_c^{(1)} = D([\hat{f}_{shape}; f_{style}])$. Symmetrically, Branch 2 processes $P_o^{(2)}$ through identical modules to produce $P_c^{(2)}$. Implementation specifics of Feature Disentanglement Module and PPM are delineated as follows:

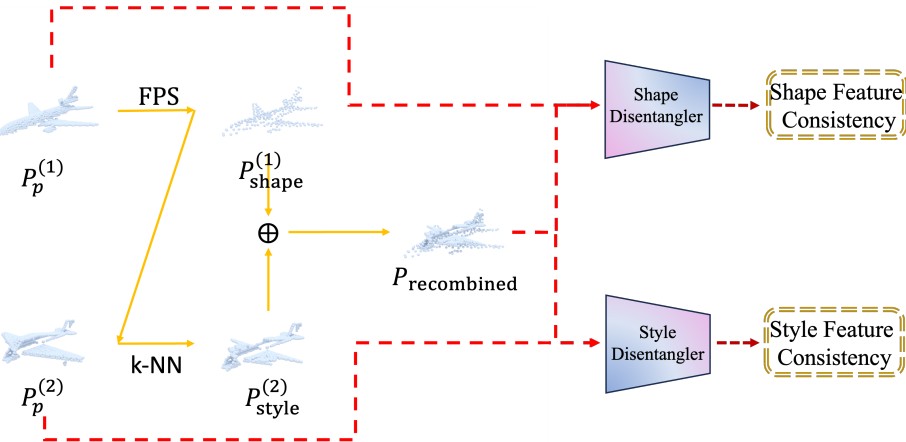

Figure 3: Illustration of Feature Permutation Consistency Constraint (FPCC). (a) Structural points $P_{\text{shape}}^{(1)}$ are extracted from input cloud $P_p^{(1)}$ via farthest point sampling (FPS). (b) The $k$-nearest neighbors from $P_p^{(2)}$ are aggregated as style points $P_{\text{style}}^{(2)}$. (c) Recombined point cloud $P_{\text{recombined}}$ is generated by concatenating $P_{\text{shape}}^{(1)}$ and $P_{\text{style}}^{(2)}$, which preserves the global structure of $P_p^{(1)}$ and local details of $P_p^{(2)}$. (d) FPCC enforces feature consistency between $P_{\text{recombined}}$ and original clouds for effective feature disentanglement.

**Feature Disentanglement Module.**    Within the feature disentanglement module, we decompose point cloud features into shape features representing global structures and style features capturing local details. We can effectively aggregate complete structures through cluster-level contrastive learning to the shape features avoiding the interference from local detail variations. And the disentangled style feature can be also used as to preserve instance-specific details details of input. To ensure the disentangled features correspond accurately to global structure and local details, respectively, we propose a **Feature Permutation Consistency Constraint (FPCC)**, as illustrated in Figure 3. Specifically, we randomly select two incomplete point clouds, $P_p^{(1)}$ and $P_p^{(2)}$. Then, we perform FPS on $P_p^{(1)}$ to obtain its structural points, $P_{\text{shape}}^{(1)}$. For each point in $P_{\text{shape}}^{(1)}$, we find $k$-nearest neighbors in $P_p^{(2)}$, forming $P_{\text{style}}^{(2)}$. By concatenating $P_{\text{shape}}^{(1)}$ and $P_{\text{style}}^{(2)}$, we construct the recombined point cloud $P_{\text{recombined}}$. We assume that $P_{\text{recombined}}$ contains the global structure of $P_p^{(1)}$ and the local details of $P_p^{(2)}$, so we require their corresponding features to remain consistent, which is formulated as:

$$\mathcal{L}_{\text{fpcc}} = \phi\left(f_{\text{shape}}^{(1)} - f_{\text{shape}}^{\text{recombined}}\right) + \phi\left(f_{\text{style}}^{(2)} - f_{\text{style}}^{\text{recombined}}\right), \tag{1}$$

where $f_{\text{shape}}^{(1)}$, $f_{\text{shape}}^{\text{recombined}}$, $f_{\text{style}}^{(2)}$, and $f_{\text{style}}^{\text{recombined}}$ are the features extracted from the corresponding point clouds through the network, and $\phi(\cdot)$ is the Huber loss function [18]. Through FPCC, we effectively disentangle the shape and style features of the point cloud.

**Prototype Projection Module (PPM).**    Specifically, in the PPM, we maintain a learnable prototype memory bank $\mathcal{M}$. With training (see Section 3.2 for details), each item $\mathcal{M}_i \in \mathbb{R}^{D/2}$ is considered as a prototype representing to a unique complete structure. For the input shape feature $f_{\text{shape}}$, we compute its cosine similarity with each prototype and apply a softmax function to obtain a weight matrix $W$. The computation formula is as follows:

$$w_i = \frac{\exp\left(\cos(f_{\text{shape}}, \mathcal{M}_i/t)\right)}{\sum_{j \in |\mathcal{M}|} \exp\left(\cos(f_{\text{shape}}, \mathcal{M}_j/t)\right)}, \tag{2}$$

where $t$ is a temperature scaling parameter. Then, based on the weight matrix $W$, we project $f_{\text{shape}}$ into the prototypes to obtain the refined feature $\hat{f}_{\text{shape}}$, i.e., $\hat{f}_{\text{shape}} = \sum_{i \in |\mathcal{M}|} w_i \cdot \mathcal{M}_i$.

## 3.2 Cluster-level Contrastive Learning

We employ cluster-level contrastive learning to perform clustering on shape features and utilize the learned cluster centers as complete structural information to guide point cloud completion. Specifically, we maintain a learnable prototype memory bank $\mathcal{M}$, where each item $\mathcal{M}_i \in \mathbb{R}^{D/2}$ represents the cluster center with aggregated complete structure and is random initialized. We then define a mapping function $\psi$, which maps a point cloud $P_p$ to its nearest shape prototype $\mathcal{M}_i$, i.e., $\psi(P_p) = \mathcal{M}_i$, where $i = \arg\max_j \cos(D_{\text{shape}}(E(P_p), \mathcal{M}_j))$.

We propose an aggregation loss $\mathcal{L}_{\text{agg}}$ to learning shape prototypes. The implementation pipeline contains three key phases: 1) The network first produces the completed point cloud $P_c$ from the partial input $P_p$. 2) Then, we concatenate the shape prototype $\mathcal{M}_i$ with a zero vector and input it into the decoder to generate the complete structural point cloud $P_c^{\mathcal{M}_i}$. 3) We calculate Chamfer Distance (CD) between the cluster-specific structural point cloud $P_c^{\mathcal{M}_i}$ and predicted complete point cloud $P_c$, which is formulated in Eq.3:

$$\mathcal{L}_{\text{agg}} = CD(P_c, P_c^{\mathcal{M}_i}), \tag{3}$$

where CD takes both directions into account and can be defined through Unidirectional Chamfer Distance(UCD) as $CD(P_1, P_2) = UCD(P_1, P_2) + UCD(P_2, P_1)$. The UCD is commonly used to measure the distance between two point clouds [38]. UCD between two point sets $S_1$ and $S_2$ is defined as follows:

$$UCD(S_1, S_2) = \frac{1}{|S_1|} \sum_{x \in S_1} \min_{y \in S_2} \|x - y\|_2. \tag{4}$$

In addition, we introduce an InfoNCE loss [16] to ensure that the learned shape prototypes maintain sufficient discriminability for preventing prototype collapse. This loss encourages the shape feature of the point cloud to be as close as possible to the mapped prototype, while keeping far away from others. The InfoNCE loss is formulated as:

$$\mathcal{L}_{\text{Info}} = -\log \frac{\exp\left(\cos(f_{\text{shape}}, \psi(P_p)))/t\right)}{\sum_{i=1}^{|\mathcal{M}|} \exp\left(\cos(f_{\text{shape}}, \mathcal{M}_i))/t\right)}, \tag{5}$$

where $f_{\text{shape}} = D_{\text{shape}}(E(P_p))$. Thus, the final loss for the cluster-level contrastive learning is defined as below:

$$\mathcal{L}_{\text{cl}} = \mathcal{L}_{\text{agg}} + \mathcal{L}_{\text{Info}}. \tag{6}$$

## 3.3 Instance-level Contrastive Learning

We propose using instance-level contrastive learning to ensure that the extracted style features focus on the details of the input point cloud. Specifically, as shown in Figure 2, two branches are employed during training, where the inputs to the branches, $P_o^{(1)}$ and $P_o^{(2)}$, are obtained by applying different masks to the same sample. $P_o^{(1)}$ and $P_o^{(2)}$ serve as positive samples in the context of contrastive learning. By enforcing the two branches to produce identical completed point clouds, the style features are guided to capture the unique details of the input. The instance-level contrastive learning loss is formulated as follows:

$$\mathcal{L}_{\text{instance}} = CD(P_c^{(1)}, P_c^{(2)}). \tag{7}$$

## 3.4 Optimization

Finally, we introduce a completion loss to for point cloud completion. By computing the Region-Aware Chamfer Distance (RCD) [7] between $P_p$ and the completed point clouds $P_c^{(1)}$ and $P_c^{(2)}$, the network is encouraged to reconstruct the visible regions while allowing the completion of unseen regions. The formulation is as follows:

$$\mathcal{L}_{\text{completion}} = RCD(P_p, P_c^{(1)}) + RCD(P_p, P_c^{(2)}). \tag{8}$$

where RCD is aware of observed and unseen regions and thus only evaluates point distance for observed regions, we refer the reader to [7] for more details about RCD. Every input mapped to shape prototype $\mathcal{M}_i$ participates in the aggregation loss computation, thereby enabling $\mathcal{M}_i$ to aggregate the

Table 1: Quantitative comparison result of our method and other methods on the 3D-EPN dataset using CD-$\ell_2 \downarrow (\times 10^4)$. Bold numbers indicate the best performance in self-supervised methods.

| Method | Supervision | Air | Cab | Car | Cha | Lam | Sof | Tab | Wat | Avg |
|--------|-------------|-----|-----|-----|-----|-----|-----|-----|-----|-----|
| PCN [38] | | 2.5 | 8.0 | 4.8 | 9.0 | 12.2 | 8.1 | 8.9 | 6.0 | 7.4 |
| TopNet [29] | Supervised | 2.3 | 7.5 | 4.6 | 7.6 | 8.9 | 7.3 | 7.5 | 5.2 | 6.4 |
| PoinTr [37] | | 1.2 | 6.5 | 4.0 | 5.1 | 4.5 | 5.4 | 5.4 | 2.6 | 4.3 |
| CRA-PCN [27] | | 0.9 | 5.9 | 3.3 | 4.2 | 3.9 | 5.5 | 3.6 | 2.3 | 3.7 |
| C4C [31] | Unpaired | 3.7 | 12.6 | 8.1 | 14.6 | 18.2 | 26.2 | 22.5 | 8.7 | 14.3 |
| Inv [39] | | 4.3 | 20.7 | 11.9 | 20.6 | 25.9 | 54.8 | 38.0 | 12.8 | 23.6 |
| Gu et al. [13] | Weakly-supervised | 5.9 | 20.8 | 9.5 | 20.4 | 34.9 | 26.0 | 26.0 | 11.0 | 21.3 |
| PPNet [21] | | 5.6 | 46.6 | 22.4 | 24.3 | 46.1 | 36.4 | 28.4 | 15.0 | 28.1 |
| ACL-SPC [17] | | 14.6 | 25.3 | 16.4 | 45.0 | 60.1 | 35.6 | 40.8 | 29.2 | 31.6 |
| P2C [7] | Self-supervised | 4.3 | 19.4 | 8.6 | 13.5 | 16.3 | 20.2 | 18.1 | 12.0 | 14.1 |
| CSG-PCC(Ours) | | **3.5** | **16.1** | **8.5** | **12.1** | **14.6** | **16.4** | **17.4** | **9.1** | **12.2** |

structural information from these partial point clouds. This can be interpreted as guiding the learning of shape prototypes via positive sample in contrastive learning.

Finally, we sum the previously introduced loss terms to obtain the total loss used for network training:

$$\mathcal{L}_{\text{overall}} = \lambda_{\text{completion}}\mathcal{L}_{\text{completion}} + \lambda_{\text{cl}}\mathcal{L}_{\text{cl}} + \lambda_{\text{instance}}\mathcal{L}_{\text{instance}} + \lambda_{\text{fpcc}}\mathcal{L}_{\text{fpcc}}, \qquad (9)$$

where $\lambda_{\text{completion}}$, $\lambda_{\text{cl}}$, $\lambda_{\text{instance}}$, and $\lambda_{\text{fpcc}}$ are weighting parameters.

## 4 Experiments

### 4.1 Dataset and Evaluation Metrics

**Datasets.** To conduct a comprehensive comparison, we performed experiments on both synthetic and real-world datasets. We conducted experiments on the synthetic datasets 3D-EPN [9] and PCN [38]. Both of them are derived from the ShapeNet [2] dataset, with the former containing more data, e.g., the chair class has 40,000 pairs for training in 3D-EPN while 5,750 pairs in PCN. Moreover, we evaluate our method on real-world dataset ScanNet [8]. We employ the ScanNet dataset processed by [5], where both object orientations and positions have been aligned with ShapeNet.

**Evaluation Metric.** We use $\ell_2$ Chamfer Distance (CD)as the evaluation metric for synthetic datasets. In the case of real-world datasets, where ground-truth complete shapes are unavailable, we adopt the Unidirectional Chamfer Distance (UCD), Unidirectional Hausdorff Distance (UHD), and Region-Aware Chamfer Distance (RCD) proposed in P2C [7] to evaluate the fidelity of completed point clouds.

### 4.2 Evaluation on Synthetic Datasets

We compare our proposed method CSG-PCC with state-of-the-art self-supervised method and classical supervised, unpaired methods on 3D-EPN datasets. As quantitatively demonstrated in Table 1, our method establishes new performance records across all eight categories when compared to existing self-supervised approaches. Furthermore, as shown in in Table 2, our methods achieve more improvements on PCN dataset. This occurs because the PCN dataset contains less training data and more severe incompleteness (lower average points in input clouds), while our method explicitly extracts complete structural information to better handle challenging scenarios. Although fully supervised methods still show numerical advantages by heavily exploiting complete and paired ground-truth data, our method has further reduced the performance gap.

Figure 4 presents a qualitative comparison on the 3D-EPN dataset. It can be observed that our method successfully completes the incomplete point clouds with guidance of complete structures. In particular, compared with P2C, our method achieves more complete results on car and sofa categories

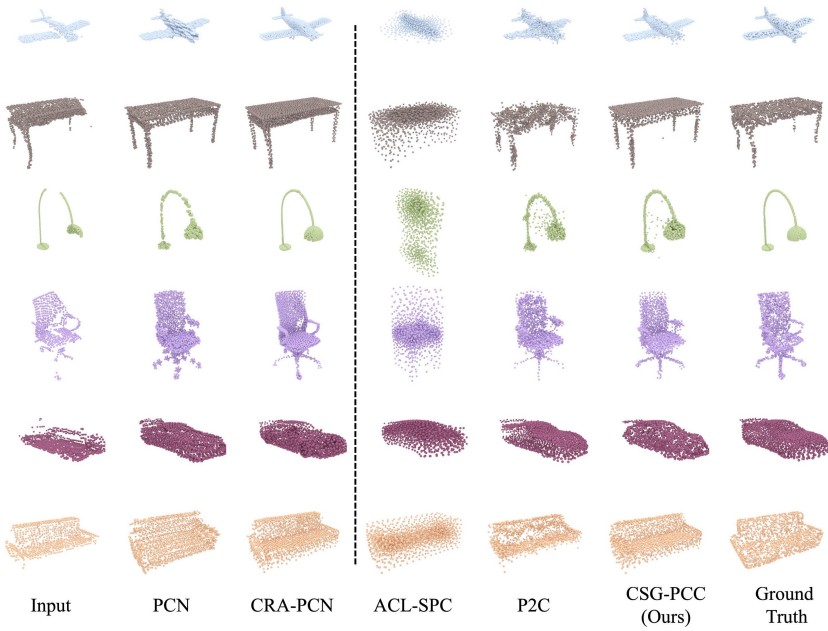

| Input | PCN | CRA-PCN | ACL-SPC | P2C | CSG-PCC (Ours) | Ground Truth |

Figure 4: Qualitative comparisons on the 3D-EPN dataset demonstrate our method achieves both *structural completeness* (e.g.,*car*, *sofa*) and *detail preservation* (e.g., *chair*, *table*). The dashed line categorizes the methods based on whether they utilize complete point clouds for supervision.

Table 2: Quantitative comparison result of our method and other methods on the PCN dataset using CD-$\ell_2 \downarrow (\times 10^4)$. Bold numbers indicate the best performance in self-supervised methods.

| Method | Supervision | Air | Cab | Car | Cha | Lam | Sof | Tab | Wat | Avg |
|---|---|---|---|---|---|---|---|---|---|---|
| PCN [38] | | 3.0 | 7.5 | 5.7 | 9.7 | 9.2 | 9.5 | 9.2 | 6.2 | 7.5 |
| TopNet [29] | Supervised | 2.3 | 8.2 | 4.7 | 8.6 | 11.0 | 9.3 | 7.5 | 5.2 | 6.4 |
| CRA-PCN [27] | | 1.1 | 6.3 | 3.8 | 4.3 | 3.4 | 5.9 | 3.7 | 2.8 | 3.9 |
| C4C [31] | Unpaired | 4.1 | 14.2 | 9.9 | 14.6 | 19.2 | 27.8 | 8.4 | 7.4 | 14.4 |
| Inv [39] | | 3.9 | 17.4 | 11.0 | 13.8 | 14.2 | 23.0 | 9.7 | 6.7 | 14.1 |
| ACL-SPC [17] | | 10.6 | 38.8 | 30.5 | 41.4 | 63.5 | 33.2 | 42.8 | 29.2 | 39.2 |
| P2C [7] | Self-supervised | 4.8 | 32.3 | 17.8 | 18.1 | 18.6 | 33.0 | 19.8 | 13.7 | 19.8 |
| CSG-PCC(Ours) | | **3.6** | **18.2** | **15.7** | **13.7** | **15.8** | **28.6** | **19.3** | **12.4** | **15.9** |

with severe incompleteness, while preserving richer details for structurally complex lamp and chair categories. We also present the qualitative comparison results on PCN data in section A.2.

## 4.3 Evaluation on Real-world Datasets

We compare our method with other approaches on real-world datasets, where our method achieves the best performance among self-supervised methods. Both PCN [38] and Inv [39] are pre-trained on the ShapeNet dataset. Inv's superior UCD metric performance over ours stems from its optimization-based nature, which explicitly employs UCD as the loss function during inference phase. However, visualization results in Figure 5 reveal that our method produces outputs that are both structurally complete and more faithful to the input.

## 4.4 Ablation Study

**Model Design Analysis.** To examine the effectiveness of our design, we conduct detailed ablation experiments on the cabinet, chair, lamp, and sofa categories of the PCN dataset, with results summa-

Table 3: Quantitative comparison result of our method and other methods on the ScanNet dataset using UCD-$\ell_2$ ↓ ($\times 10^4$), UHD↓ ($\times 10^2$), RCD↓ ($\times 10^3$). Bold numbers indicate the best performance in self-supervised methods.

| Method | UCD | | UHD | | RCD | |
|---|---|---|---|---|---|---|
| category | chair | table | chair | table | chair | table |
| PCN [38] | 5.1 | 4.9 | 6.2 | 6.0 | 2.1 | 1.8 |
| Inv [39] | 2.8 | 3.4 | 9.9 | 12.2 | 0.6 | 0.8 |
| ACL-SPC [17] | 5.7 | 6.8 | 7.0 | 7.4 | 2.3 | 3.1 |
| P2C [7] | 3.7 | 4.6 | 6.5 | 6.7 | 1.0 | 1.3 |
| CSG-PCC(Ours) | **3.3** | **3.7** | **6.1** | **6.4** | **0.8** | **0.9** |

Table 4: Ablation study on the PCN dataset. Results reported in CD-$\ell_2$ ↓ scaled by $10^4$.

| Model | FPCC | $\mathcal{L}_{cluster}$ | $\mathcal{L}_{instance}$ | Cab | Cha | Lam | Sof |
|---|---|---|---|---|---|---|---|
| A | | | | 32.3 | 18.1 | 18.6 | 33.0 |
| B | ✓ | | | 29.6 | 17.9 | 17.8 | 32.2 |
| C | ✓ | ✓ | | 19.3 | 14.8 | 16.3 | 29.4 |
| D | ✓ | ✓ | ✓ | **18.2** | **13.7** | **15.8** | **28.6** |

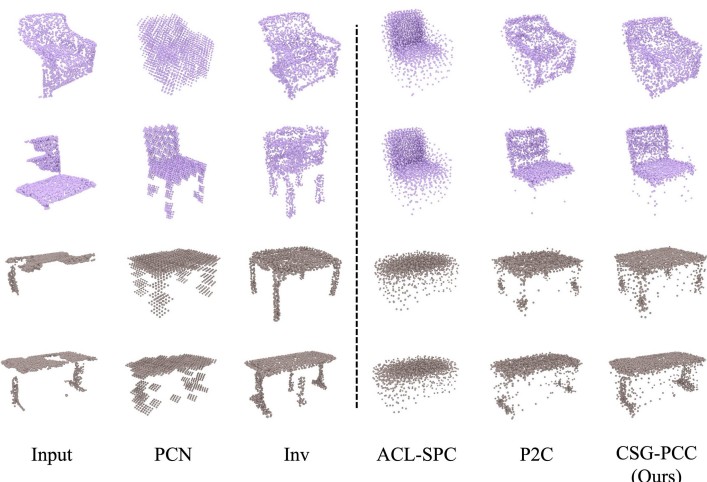

| Input | PCN | Inv | ACL-SPC | P2C | CSG-PCC (Ours) |

Figure 5: Qualitative comparisons on the ScanNet dataset. The dashed line categorizes the methods based on whether they utilize complete point clouds for supervision.

rized in Table 4. The baseline model (Model A) adopts the P2C architecture. We then modify the network by adding shape-style disentanglers and computing FPCC for feature decoupling, forming Model B. Through shape and style feature disentanglement, the network acquires awareness of global structures and local details, achieving moderate improvements across all four categories. Subsequent integration of the prototype projection module with cluster-level contrastive learning constitutes Model C, enabling explicit extraction of complete point cloud structures from shape features. Guided by this complete structural information, our method demonstrates significant performance gains, particularly benefiting severely incomplete categories like cabinets and sofas. Finally, we incorporate instance-level contrastive learning to establish our complete CSG-PCC framework (Model D), achieving state-of-the-art results. Due to space constraints, we include additional ablation experiments in the Appendix section, covering hyperparameter selection, prototype visualization, and other analyses.

## 5 Limitations

Although CSG-PCC has demonstrated promising results in completing point clouds with only single partial data needed for learning, several limitations still need to be addressed. Notably, all structural information in our method is derived solely from the training data. If the incomplete point clouds in the training set inherently lack certain structures (e.g., all aircraft samples missing wings), our approach cannot recover such missing components (e.g., wings). This limitation aligns with existing self-supervised methods.

## 6 Conclusion

In this paper, we propose a noval self-supervised point cloud completion framework, CSG-PCC, which leverages self-learned complete structures to guide the completion process. Our method achieves the extraction of complete structures and local details from incomplete point clouds through feature disentanglement and dual-level contrastive learning. Experiments demonstrate that the point clouds completed by our method are structure-completed and detail-preserving, and exhibit state-of-the-art performance of self-supervised methods on both synthetic and real-world datasets.

## Acknowledgments and Disclosure of Funding

This work was supported by the National Natural Science Foundation of China (Grant No. 62176064, NO. 62472097), and AI for Science Foundation of Fudan University (FudanX24AI028).

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

# A   Technical Appendices and Supplementary Material

## A.1   Implementation Details

We use the encoder from PCN [38] and employ an MLP with two hidden layer as our decoder. Similar to OptDE [11], we take two separate disentanglers, each implemented as MLPs with one hidden layer, to extract shape and style features. For the loss function, we set the $\lambda_{\text{completion}}$, $\lambda_{\text{cluster}}$, $\lambda_{\text{instance}}$ to 1, 0.1, and 0.01, respectively. The FPCC is computed every 50 backpropagation steps with a weight of 0.1. Like P2C [7], we divide the incomplete point cloud into 64 patches, each containing 32 points. The num of shape prototypes $|\mathcal{M}|$ is set to 32 and the temperature scaling parameter $t$ is set to 0.4. We train a separate network for each class using the AdamW optimizer with a starting learning rate of $10^{-3}$ and a weight decay of $10^{-3}$ for 300 epochs. The experiments were conducted on four NVIDIA GeForce RTX 3090 GPUs with 24GB memory each.

## A.2   Qualitative comparisons on the PCN dataset

We also present qualitative comparisons in Figure 6 on the PCN dataset. When trained with less data and more incomplete inputs, our method outperform existing self-supervised approaches in both completion quality and input faithfulness, corroborating the robustness of our approach.

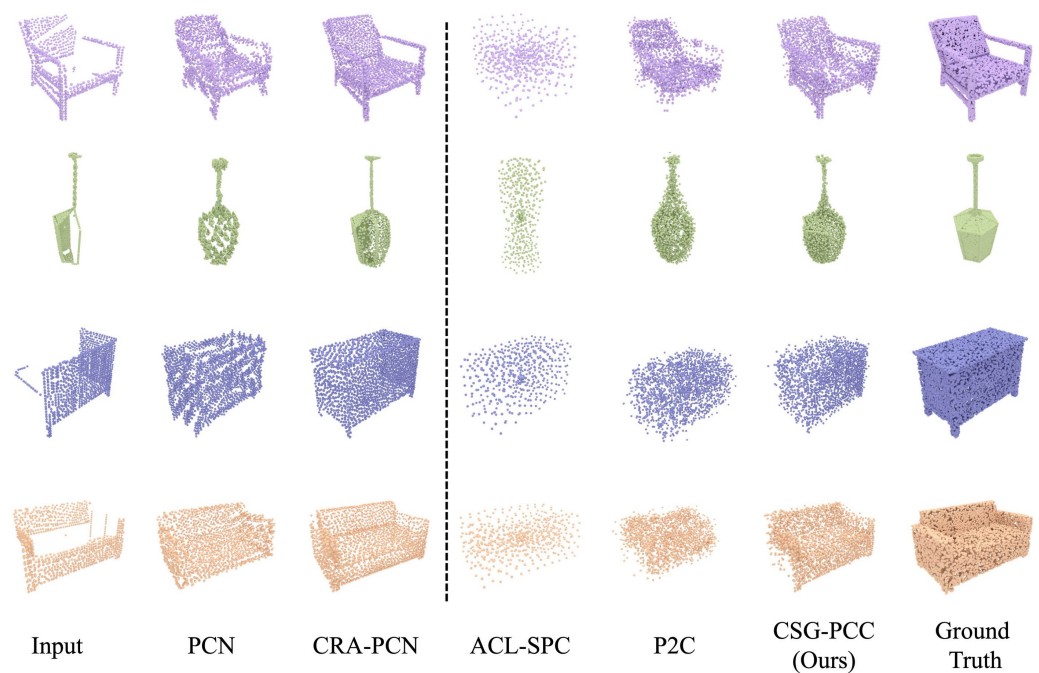

| Input | PCN | CRA-PCN | ACL-SPC | P2C | CSG-PCC (Ours) | Ground Truth |

Figure 6: Qualitative comparisons on the PCN dataset. The dashed line categorizes the methods based on whether they utilize complete point clouds for supervision.

## A.3   Visualization of prototypes.

We concatenate the learned shape prototypes with zero vectors and pass them through the decoder to obtain the point clouds corresponding to the prototypes. By visualizing these point clouds, as shown in Figure 7, we can observe that the learned shape prototypes are structurally complete.

## A.4   Hyperparameter Selection.

We conduct an empirical study to investigate the impact of hyperparameters.

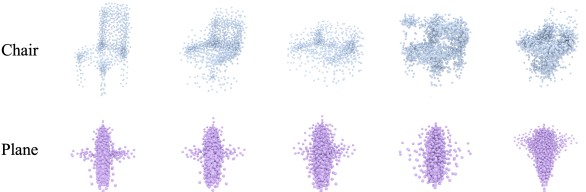

Figure 7: Visualization of prototypes.

Table 5: The effect of number of shape prototypes. Results reported in CD-$\ell_2$ $\downarrow$ scaled by $10^4$.

| Category | 0.1 | 0.4 | 0.7 |
|---|---|---|---|
| Cab | 21.9 | **18.2** | 19.6 |
| Cha | 15.1 | **13.7** | 14.4 |
| Lam | 17.5 | **15.8** | 16.2 |
| Sof | 30.2 | **28.6** | 29.1 |
| Avg | 20.2 | **19.1** | 19.8 |

**Impact of the number of shape prototypes.** In particular, we examine the impact of the number of shape prototypes on the model's performance. The results show in Table 5. We observe that the optimal number of prototypes varies across different categories. However, we set the prototype number to 32 for all categories for simplicity, yielding results superior to existing self-supervised methods.

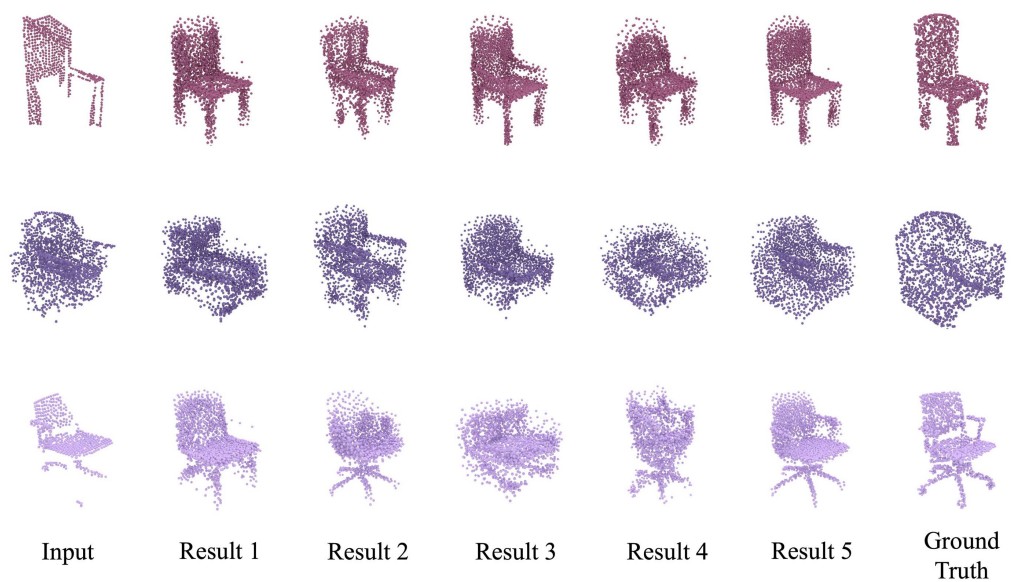

| Input | Result 1 | Result 2 | Result 3 | Result 4 | Result 5 | Ground Truth |
|---|---|---|---|---|---|---|

Figure 8: Visualization of multimodal point cloud completion.

**impact of the temperature parameter.** Additionally, we investigate the impact of the temperature parameter $t$ in the prototype projection module on performance in Table 6. We test three different values of $t$ across four categories, observing that the network achieves optimal performance when $t = 0.4$. Higher $t$ values cause all negative samples to be treated uniformly with lacking discrimination, whereas lower $t$ values lead to insufficient reference to similar complete structures.

Table 6: The effect of temperature values. Results reported in CD-$\ell_2$ $\downarrow$ scaled by $10^4$.

| Category | 8 | 16 | 32 | 64 |
|----------|------|------|------|------|
| Cab | 18.7 | **17.8** | 18.2 | 21.4 |
| Cha | 14.8 | 14.5 | **13.7** | 13.9 |
| Lam | 16.9 | 16.4 | 15.8 | **14.9** |
| Sof | 29.7 | **28.1** | 28.6 | 29.7 |
| Avg | 20.3 | 19.2 | **19.1** | 20.0 |

Table 7: Complexity and efficiency analysis in terms of the number of parameters (Params) and frames per second (fps) with the average Chamfer Distance CD-$\ell_2$ $\downarrow$ on the 3D-EPN dataset as references.

| Method | Params$\downarrow$ | fps$\uparrow$ | Avg.CD-$\ell_2$ |
|--------|--------|--------|--------|
| PCN [38] | 4.1M | 20.6 | 7.4 |
| CRA-PCN [27] | 22.3M | 14.7 | 3.7 |
| Inv [39] | 40.1M | 0.03 | 23.6 |
| ACL-SPC [17] | 8.1M | 17.4 | 31.6 |
| P2C [7] | 23.9M | 21.3 | 14.1 |
| CSG-PCC(Ours) | 27.1M | 19.5 | 12.2 |

## A.5 Complexity and Efficiency Analysis.

Additionally, we conduct complexity and efficiency analyses in Table 7. While our method entails a moderate parameter increase (due to two disentanglers and prototype memory bank) and slight inference speed reduction compared to P2C, the resultant performance gains justify this trade-off.

## A.6 Multimodal Point Cloud Completion.

Furthermore, due to the design of shape prototypes, we can concatenate the style feature with different shape prototypes to generate different completed point clouds, i.e., multimodal point cloud completion [33]. The multimodal point cloud completion results are shown in Figure 8. It can be observed that our method produces diverse yet plausible completion results for single incomplete input, which demonstrates potential of our methods for multimodal point cloud completion tasks.

