# OpenReview forum: "Complete Structure Guided Point Cloud Completion via Cluster- and Instance-Level Contrastive Learning"
_NeurIPS.cc/2025/Conference — NeurIPS 2025 spotlight_

### Official Review · Reviewer_CKyN · 2025-06-30

**Clarity:** 3
**Significance:** 3
**Originality:** 3
**Rating:** 4
**Confidence:** 5

**Summary:**

In this work, the authors propose a method to train a point cloud completion network without GT completed point clouds for supervision. During training, the input point cloud is firstly repeatedly corrupted into two homologous but different point clouds processed with identical completion network, whose completed results would be used to calculated instance-level contrastive loss.
In the completion network, each corrupted input point cloud is disentangled into two branches: the shape branch and style branch. The shape branch extracts shape features from the FPS sampled points, and completed them by "clustering" the 3D partial points in the training set with a memory bank, where the style branch help preserves details from existing partial geometry.

**Questions:**

Please check the weaknesses section, thanks.

**Ethical Concerns:**

["NO or VERY MINOR ethics concerns only"]

**Final Justification:**

Thank you to the authors for their response. After reading the rebuttal, I remain generally positive about the work. I would also strongly encourage the authors to clearly address the mentioned concerns in the revised version to further strengthen the paper. Thank you.

**Limitations:**

Yes, in the appendix.

**Quality:**

2

**Strengths And Weaknesses:**

Strengths:

1. The proposed method does not rely on complete 3D ground truth for training. Instead, it leverages diverse observations aggregated from the training set, reducing the need for extensive labeled data.

2. The method demonstrates superior performance on both synthetic datasets (3D-EPN, PCN) and real-world data (ScanNet), highlighting its effectiveness and generalizability.

Weaknesses:

1. The core of the method relies on shape clustering across the training set to populate a memory bank with shared features. However, this process is heavily dependent on the diversity and scale of the dataset. For example, if only rectangular tables are present during training, the model may struggle to complete round tables or generalize to different object categories like chairs.

2. The motivation for incorporating instance-level contrastive learning remains unclear and would benefit from a more thorough explanation. Clarifying its role and contribution to performance would strengthen the paper.

3. Except the mentioned methods in the main paper, there are some other papers working on zero-shot point cloud completion based on 2D pre-trained diffusion priors, which also do not require any training on completed data, such as [1, 2]. Some comparisons or discussions would be essential.

[1] Point cloud completion with pretrained text-to-image diffusion models

[2] ComPC: Completing a 3D Point Cloud with 2D Diffusion Priors

4. The paper could be further strengthened by including more comparisons on datasets that lack ground truth, such as KITTI. This would better demonstrate the practical utility of the method in real-world scenarios.

5. The manuscript would benefit from additional proofreading and editing for clarity and accuracy. For instance:

(1) Line 176 refers to "RCD in Eq. 3," but it appears to be standard Chamfer Distance instead.

(2) The term "UCD" is used without definition—please clarify if this refers to Uniform Chamfer Distance.

(3) In the caption of Figure 5, "scanNet" is incorrectly written and should be corrected.

---

> ### Author Rebuttal · Authors · 2025-07-31
>
> We greatly appreciate your positive feedback and thoughtful suggestions regarding our paper. In particular, the paper you recommended provides a feasible direction for the future improvement of our method. Furthermore, your comments on both the content and the experiments have helped us further enhance the clarity and persuasiveness of our paper. Next, we will respond to your suggestions point by point.
>
> **Weakness 1:** The core of the method relies on shape clustering across the training set to populate a memory bank with shared features. However, this process is heavily dependent on the diversity and scale of the dataset. For example, if only rectangular tables are present during training, the model may struggle to complete round tables or generalize to different object categories like chairs.
>
> Yes, we have acknowledged this limitation in our paper in Appendix.7. Our method is indeed constrained by the diversity and scale of the training data. However, this is a common challenge faced by current self-supervised point cloud completion methods. Nevertheless, the significance of self-supervised approaches lies in the fact that we can collect incomplete point clouds as training data and then perform completion, without the need for complete point clouds as supervision. Besides, in such scenarios (e.g., when the training data lacks diversity in specific shapes or categories), our method can mitigate the risk of being misled by clustered prototypes by simply skipping the PPM. Meanwhile, the instance-level contrastive learning component remains effective, ensuring reasonable completion performance.
>
> Considering the methods you recommended (based on 2D pre-trained diffusion priors), we agree that leveraging powerful 2D priors or cross-modal pre-training strategies could be a promising way to alleviate the reliance on extensive 3D training data. In response to your suggestion, we will add a discussion in the appendix (see **Weakness 3** ) to highlight this potential improvement, and outline how future research could benefit from such approaches.
>
> **Weakness 2:** The motivation for incorporating instance-level contrastive learning remains unclear and would benefit from a more thorough explanation. Clarifying its role and contribution to performance would strengthen the paper.
>
> Thank you for highlighting the need for a clearer motivation regarding the incorporation of instance-level contrastive learning. We address your concerns as follows:
>
> * **Clarification of Role:** In our pipeline, i.e., Figure 2, the two incomplete point clouds in the dual-branch network are generated from a single input instance. Consequently, the completed results from both branches should ideally be consistent. By encouraging the network to minimize the discrepancy between these two outputs, i.e., $\cal L_{instance}$ shown in Eq.(7). We treat them as positive pairs in a contrastive sense, thereby promoting consistency and preservation of instance-specific details.
> * **Contribution to Performance:** The effect of the proposed instance-level contrastive loss ($ \cal L_{instance}$ ) is demonstrated in our ablation study (Section 4.4). As shown in Table 4, adding $\cal L_{instance}$ leads to more significant improvements in categories with richer details, such as lamp and chair, compared to cabinet and sofa. This supports our claim that instance-level contrastive learning is particularly beneficial for retaining local details in the completion results.
>
> **Weakness 3:** Except the mentioned methods in the main paper, there are some other papers working on zero-shot point cloud completion based on 2D pre-trained diffusion priors, which also do not require any training on completed data, such as [1, 2]. Some comparisons or discussions would be essential.
>
> [1] Point cloud completion with pretrained text-to-image diffusion models
>
> [2] ComPC: Completing a 3D Point Cloud with 2D Diffusion Prior
>
> Thank you for pointing out these two outstanding works, which are indeed highly valuable references. Their utilization of 2D pre-trained diffusion priors effectively reduces the reliance on large-scale 3D training datasets. However, it is important to note that these two methods belong to a different branch of point cloud completion from our method together with the baselines in this paper.  So a direct comparison with our method may not be entirely fair, particularly in terms of efficiency. Both [1] and [2] employ optimization-based algorithms, which perform extensive optimization during the completion process for a given point cloud. In contrast, our method together with the baselines learn a completion network during training and only perform a single forward pass in completing each point cloud. As reported in [2], completing a single sample requires 15 minutes on an A6000 GPU, while [1] requires up to 1950 minutes per sample. In contrast, our method completes 19.5 samples per second on an RTX 3090 GPU. Even without considering hardware differences, this means our method is orders of magnitude faster than [2]. Such discrepancy in efficiency between [1-2] and our method implies that they would be used in different applications with differenet requirements on latency.
>
> Furthermore, we believe that methods [1–2] can be combined with our method to improve efficiency, where our method can provide a coarse completion result as input to optimization-based methods such as [2], which can then finetune the final results.
>
> We will incorporate the above discussion into the revised version of our paper to provide a more comprehensive and balanced perspective.
>
> **Weakness 4:** The paper could be further strengthened by including more comparisons on datasets that lack ground truth, such as KITTI. This would better demonstrate the practical utility of the method in real-world scenarios.
> Thank you for your constructive suggestion.  As recommended, we have conducted experiments on the car category of the KITTI dataset processed by the pcl2pcl [5]. The dataset contains 7,569 samples, with an average of 527 points per incomplete point cloud. The results are presented in the table below:
> | Method | Supervision | UCD  | UHD  | RCD  |
> |------|-----|------|------|------|
> | PCN[38]        | Supervised | 37.3 | 14.0 | 6.57 |
> | Inv [39]       |  Unpaired  | 3.16 |   7.11  |  1.34 |
> | ACL-SPC [17]   | Self-supervised | 9.75 |  6.89  |  3.95 |
> | P2C [7]        | Self-supervised |3.61 | 6.98 | 1.22 |
> | CSG-PCC(Ours)  |     Self-supervised    |3.38 | 6.17 | 1.21 |
>
> As shown, our method outperforms existing self-supervised point cloud completion methods on the KITTI dataset. The superior performance of Inv[39] on the UCD metric compared to our method stems from its optimization-based nature, as it explicitly employs the UCD loss during inference, as mentioned in the main text on lines 239–240. Regarding supervised methods, our approach achieves a notably greater improvement on KITTI compared to our experiments on the ScanNet dataset as shown in Table 3. This is because, for ScanNet, we used only 550 chairs and 552 tables for training and evaluation, while for KITTI, we utilized 7,569 cars. As the amount of training data increases, the domain gap becomes more pronounced for supervised methods. That is, as we discussed in line 31 to 32 in the main text, they tend to suffer from limited generalization ability for they are trained on virtual data and tested on real-world data. In contrast, our method can be directly trained on real incomplete point clouds, thereby mitigating the impact of domain differences. This further demonstrates the practical utility and robustness of our approach in real-world settings.
>
> We will include these experimental results and analyses in the revised version of the paper.
>
> **Weakness 5:**  The manuscript would benefit from additional proofreading and editing for clarity and accuracy. For instance:
> (1) Line 176 refers to "RCD in Eq. 3," but it appears to be standard Chamfer Distance instead.
> (2) The term "UCD" is used without definition—please clarify if this refers to Uniform Chamfer Distance.
> (3) In the caption of Figure 5, "scanNet" is incorrectly written and should be corrected.
>
> We will thoroughly proofread the manuscript and correct all typos. We appreciate your attention to details, which helps to improve the quality of our paper.

---

> ### Author Response · Authors · 2025-08-03
>
> Dear Reviewer CKyN,
>
> We hope our responses have sufficiently addressed all your inquiries. Please feel free to let us know if there remain any unanswered questions, and we would be delighted to engage in further discussion to clarify or elaborate as needed.
>
> Additionally, we have noticed that you adjusted the confidence for our paper from 4 to 5. Could we kindly ask if this change reflects any additional concerns we should address? Additionally, as we’re unable to see the updated Rating directly now, we would greatly appreciate learning the current Rating score.
>
> We remain fully committed to resolving all your concerns and welcome any additional feedback to support this process.
>
> Thank you for your time and constructive input.

---

### Official Review · Reviewer_kAqv · 2025-07-01

**Clarity:** 3
**Significance:** 3
**Originality:** 3
**Rating:** 5
**Confidence:** 5

**Summary:**

This paper addresses the self-supervised point cloud completion task. The authors identified the absence of complete structure difficulty in the point cloud data during training. They proposed a method to train the model with self learned complete structure produced from the novel self-supervised complete structure reconstruction module. Additionally, a contrastive learning approach at both the cluster and instance-level to extract shape features guided by the complete structure and to capture style features, respectively. The experiment shows the effectiveness of the proposed method.

**Questions:**

1. How was dual-level contrastive learning was illustrated in Figure 2

2. In the visual results, for example the lamp (row 2) of Figure 4, the result shows some outlier points. What could be the reason? Is this common in the results? Does this imply some design faults of this framework?

**Ethical Concerns:**

["NO or VERY MINOR ethics concerns only"]

**Final Justification:**

My final justification is maintaining my positive rating.

**Limitations:**

yes

**Quality:**

3

**Strengths And Weaknesses:**

Strengths:

1. previous self-supervised completion methods mainly employs patch-based methods to construct pseudo complete structure supervisions, while this paper proposed to create such complete structure supervision by aggregating point clouds in clusters created from contrastive learning, which is interesting and promising.

2. The design of the complete structure reconstruction is interesting, and the formulation of point cloud features as shape and style is reasonable.

3. The Feature Permutation Consistency Constraint (FPCC) is also interesting.

4. The performance and visual results look good. It shows more predicted point clouds in missing regions compared to previous methods.

Weaknesses:

1. An illustrative figure should be self-contained. However, it is difficult to clearly understand the FPCC if only read Figure 3. The authors may need to enhance the caption and content of this figure.

2.  The last paragraph of the introduction section is too long. The authors may separate the "We conducted experiments ..." as another paragraph.; The limitation section is important, should be part of the main paper.

3. typos: Figure 2, there is an additional space between Similarity and Calculation; and missing space in "b)Prototype"; Table 1 caption,  3d-EPN should be 3D-EPN

---

> ### Author Rebuttal · Authors · 2025-07-31
>
> We are honored that our work has received your recognition. Your insightful comments and constructive suggestions are extremely valuable for further enhancing our manuscript. In the following, we carefully address the weaknesses and questions you pointed out, and describe the corresponding revisions we will make.
>
> **Weakness 1:** An illustrative figure should be self-contained. However, it is difficult to clearly understand the FPCC if only read Figure 3. The authors may need to enhance the caption and content of this figure.
>
> Thank you for your constructive suggestion. In our framework, “shape” refers to the global structure of a point cloud, which is captured by the skeleton points obtained through farthest point sampling. In contrast, “detail” (or “style”) refers to the local geometric features, represented by the K-nearest neighbor points around each skeleton point. Based on this setup, we generate $P_{recombined}$ from $P_p^{(1)}$ and $P_p^{(2)}$, as shown in Figure 3. The $P_{recombined}$ has the same shape as $P_p^{(1)}$, so we use the shape disentangler in Figure 2 to extract shape features from both and ensure shape feature consistency in Eq.(1). Similarly, the $P_{recombined}$ has the same style as  $P_p^{(2)}$, so we use the style disentangler to extract style features from both and ensure style feature consistency. The combination of these two consistencies yields the FPCC loss in Eq.(1). This loss is part of the overall loss function and is used for model training, as shown in Eq.(9)
>
> To be self-contained, we will revise Figure 3 by explicitly illustrating the feature extraction process of the disentanglers, making the flow of information clearer. Additionally, we will enhance the caption to clearly state that the primary purpose of the FPCC is to guide feature disentanglement. Furthermore, in Figure 2, we will highlight the encoder and disentanglers using bounding boxes and visually indicate their relationship to the FPCC, thereby clarifying their connections within the feature disentanglement module. Through these improvements, we aim to make Figures 2 and 3 both more self-contained and mutually complementary, thereby enhancing the overall coherence and clarity of the manuscript.
>
> **Weakness 2:** The last paragraph of the introduction section is too long. The authors may separate the "We conducted experiments ..." as another paragraph.; The limitation section is important, should be part of the main paper.
>
> We will start a new paragraph from “We conducted experiments …” in the introduction section to improve readability. In addition, we will move the Limitation section to Section 5 and the Conclusion to Section 6, as you suggested.
>
> **Weakness 3:** typos: Figure 2, there is an additional space between Similarity and Calculation; and missing space in "b)Prototype"; Table 1 caption, 3d-EPN should be 3D-EPN
>
> We will thoroughly proofread the manuscript and correct all such issues. We appreciate your attention to details, which helps to improve the quality of our paper.
>
> **Q 1:** How was dual-level contrastive learning was illustrated in Figure 2
> Thank you for your question regarding the illustration of dual-level contrastive learning in Figure 2. The dual-level contrastive learning in our method refers to the instance-level and cluster-level contrastive strategies separately :
> 1. **Instance-level contrastive learning** is reflected in Figure 2 by the two parallel branches, where each branch processes an incomplete point cloud derived from the same original sample. The completion results of these two branches are expected to be consistent; therefore, we treat the outputs from the same input as positive pairs for contrastive learning, ultimately formulated as $\cal L_{instance}$ in Figure 2, whose detailed formulation is given in Eq.(7). This is visually indicated by the “Pull Close” arrow in Figure 2.
> 2. **Cluster-level contrastive learning**  is illustrated in Figure 2 by PPM. Here, the structure of the point cloud is mapped into the Prototype Memory Bank. We use contrastive learning to pull the shape features closer to their corresponding prototype, while pushing them away from other prototypes, thus capturing the complete structures at the cluster level through $\cal L_{cluster}$ in Eq.(6). This is visually indicated by the “Pull Close” and “Pull Away” arrow in Figure 2.
>
> **Q 2:** In the visual results, for example the lamp (row 2) of Figure 4, the result shows some outlier points. What could be the reason? Is this common in the results? Does this imply some design faults of this framework?
>
> Thank you for your insightful question. We have carefully analyzed the causes and have updated our explanation as follows:
> The occurrence of outlier points in the lamp category primarily arises from the intrinsic properties of this category. Compared to categories such as chairs, airplanes, or tables, lamps 1) have significantly fewer samples; 2) and much higher intra-class variation (e.g., table lamps, chandeliers, pendant lamps); 3) lamps often exhibit complex structures, low symmetry;4) in many cases, the centroid of the sample is not located within the main body of the object. These factors make the completion task for lamps particularly challenging and can lead to the appearance of outlier points in the results.
>
> The occurrence of such outlier points is not unique to our method but is also observed in previous approaches (as can be seen in Figure 4). This suggests that it is not a design fault of our framework, but rather a reflection of the challenging nature of this category and the limited data available. As discussed in the limitations section, our method, like other self-supervised point cloud completion techniques, is sensitive to the quality and diversity of the training data. Nevertheless, our prototype memory mechanism helps to cluster complete structural features, thereby reducing the negative impact of sub-category variations within the lamp class. As demonstrated in Figure 4, our approach substantially reduces the number and severity of outlier points compared to previous methods.
>
> We appreciate the reviewer’s suggestion, and we will further discuss the limitations and potential future directions for addressing outlier points in the lamp category in the revised manuscript if accepted.

---

> > ### Comment · Reviewer_kAqv · 2025-08-04
> >
> > I appreciate the authors' efforts in providing additional explanations. Most of my concerns have been addressed, and I will maintain my positive rating.
> >
> > However, when reading other reviewers' comment. I noticed that Reviewer CKyN mentioned UCD was not well-defined. It is not Uniform Chamfer Distance, but unidirectional chamfer distance. The author should clarify this.

---

> ### Author Response · Authors · 2025-08-04
>
> We sincerely appreciate your positive feedback and recognition of our efforts to address the initial concerns. It is reassuring to learn that most of your comments have been resolved, and we are glad to maintain your supportive rating.
>
> Regarding the point raised about the definition of UCD, we will explicitly provide the full term 'unidirectional Chamfer Distance' in the revised version to avoid ambiguity.
>
> Thank you again for your constructive guidance.

---

### Official Review · Reviewer_WfDZ · 2025-07-04

[review text omitted: it was posted to a different submission]

---

> ### Author Rebuttal · Authors · 2025-07-31
>
> We regret to inform you that **the review comments we have received are not related to our submission**.
> The current comments pertain to a paper on object pose optimization and object reconstruction, discussing modules such as Reconstructed Shape Supervision (RSS), Predicted Shape Guided Sampling (PSGS), and the “free lunch” claim. However, our submission focuses on self-supervised point cloud completion, and none of these modules or terms are present in our work.
>
> We have attempted to seek assistance from the AC during the rebuttal period to obtain the correct set of review comments. Unfortunately, by the end of the first phase of rebuttal, we have not received the appropriate comments.  **As a result, we are currently unable to respond meaningfully to the provided comments.** We hope you can understand the difficult situation we are in. Our lack of response to the review comments is not intentional, but rather due to the fact that the comments do not pertain to our submission.

---

> > ### Comment · Reviewer_WfDZ · 2025-08-02
> > **Reponse to the rebuttal**
> >
> > I sincerely apologize for this oversight. As I have to review multiple papers, I may have pasted the wrong review comments, and only realized this upon reading the rebuttal. Since the rebuttal period has ended, and in the interest of fairness, I have decided not to submit any further review comments and choose my score as  boardline accpt-the average score of other reviewers-to make the further discussion be fair.
> >
> >
> > Please the ACs refer to the suggestions of the other reviewers when making your decision. Thank you again, and I apologize for any inconvenience caused.

---

> ### Author Response · Authors · 2025-08-02
>
> According to reviewer WfDZ's response, since the first-round rebuttal has ended, no additional reviews were provided for fairness. Given that the average score from other reviewers is 4, he adjusted his own score to 4 to reduce his potential bias.

---

### Official Review · Reviewer_KtxY · 2025-07-04

**Clarity:** 2
**Significance:** 3
**Originality:** 3
**Rating:** 5
**Confidence:** 4

**Summary:**

The authors proposed a self-supervised method for Point Cloud Completion. This approach explicitly learns complete structures directly from incomplete point clouds, eliminating the need for complete training data. It employs cluster-level contrastive learning to extract complete-structure-guided shape features and instance-level learning to capture style details. This integrated design ensures generated point clouds achieve both shape completion and detail preservation. Extensive experiments on synthetic and real-world datasets demonstrate that the method significantly outperforms state-of-the-art self-supervised approaches.

**Questions:**

1.Figure 1 occupies disproportionate space for limited information. Clearly illustrating the distinctions between the proposed method and prior approaches in Figure 1 would enable readers to grasp this paper’s motivation more rapidly.
2. Within which component of the overall framework (Figure 2) is the Feature Permutation Consistency Constraint incorporated? The authors state they "randomly select two incomplete point clouds, Pp1 and Pp2". Does this explicitly indicate that Pp1 and Pp2 may be selected from distinct categories? If sampled from different categories, would this introduce biases or affect experimental outcomes?
3.What network architectures constitute the two disentanglers in the Feature Disentanglement Module? How can we interpret that one models shape information while the other models detail information?
4.Which component in the Pipeline corresponds to the "lgg" section mentioned in line 173? It is recommended that the PPM module be presented in a separate, more detailed figure.
5.In Section 3.3, labeling the approach employing CD loss as "Instance-level Contrastive Learning" may not be rigorously accurate. Adopting an alternative term would not undermine the novelty of this work.
6. The abbreviated loss terms (Lcom, Lins) are used throughout the paper, while Figure 2 employs full terms. This inconsistency is inappropriate. Consistency should be maintained.

**Ethical Concerns:**

["NO or VERY MINOR ethics concerns only"]

**Final Justification:**

Most of my concerns have been addressed, and I will maintain my positive rating.

**Limitations:**

yes

**Paper Formatting Concerns:**

-

**Quality:**

3

**Strengths And Weaknesses:**

The method proposed in this paper is novel. It directly relies on prototype-based contrastive learning to infer the complete structure from incomplete point clouds within training data. This is an interesting and reasonable methodology. The paper demonstrates a well-organized structure and comprehensive experiments. However, certain sections lack clarity, such as the correspondence between modules, loss functions, and the figures, specifically concerning the "Feature Disentanglement Module" and the "Lagg" parts.

---

> ### Author Rebuttal · Authors · 2025-07-31
>
> Thank you very much for your positive feedback and valuable suggestions regarding our paper. We greatly appreciate your recognition of our work. Your comments are crucial in helping us improve the clarity and overall quality of our manuscript. We carefully address your suggestions and make the necessary revisions to enhance the paper. Below, we provide point-by-point responses to your comments.
>
> **Q1:** Figure 1 occupies disproportionate space for limited information. Clearly illustrating the distinctions between the proposed method and prior approaches in Figure 1 would enable readers to grasp this paper’s motivation more rapidly.
>
> Thank you for your insightful suggestion regarding Figure 1. We plan to revise Figure 1 by spliting it two subfigures. The left subfigure  presents the pipeline of the prior methods, while the right one will expand the current figure to a self-supervised point cloud completion pipeline, which is guided by a **complete structure reconstruction module**.  This revised subfigures will use the **complete structure reconstruction module** to explicitly illustrate the distinctions between our method and prior approaches, thereby more effectively highlighting both the motivation and novelty of our approach.
>  As the rebuttal process does not permit the submission of revised figures or PDFs, we are unable to provide the updated figure at this stage. However, we will ensure that the revised figure is included in the camera-ready version if accepted.
>
> **Q2:** Within which component of the overall framework (Figure 2) is the Feature Permutation Consistency Constraint incorporated? The authors state they "randomly select two incomplete point clouds, Pp1 and Pp2". Does this explicitly indicate that Pp1 and Pp2 may be selected from distinct categories? If sampled from different categories, would this introduce biases or affect experimental outcomes?
>
> **Regarding the FPCC integration:**
>
> The Feature Permutation Consistency Constraint (FPCC) is designed to facilitate the disentanglement of shape and style features, i.e., giueding the training of shape and style disentanglers in Figure 2.  Specifically, as shown in Figure 3, $P_{recombined}$ is generated based on $P_p^{(1)}$ and $P_p^{(2)}$. It has the same shape as  $P_p^{(1)}$, so we use the shape disentangler in Figure 2 to extract shape features from both and ensure shape feature consistency by minimizing the distance of these two features (See the first term of Eq.(1) . Similarly, $P_{recombined}$ has the same style as  $P_p^{(2)}$, so we use the style disentangler in Figure 2 to extract style features from both and ensure style feature consistency by minimizing the distances of these two features in the second term of Eq.(1) . The combination of these yields the FPCC loss defined in Eq.(1). This loss is part of the overall loss function and is used for model training, as shown in Eq.(9). To clarify this in our framework, in the revised version of Figure 2, we will explicitly highlight the encoder and the disentanglers with a bounding box and use an arrow to indicate their connection to the FPCC loss. This visual enhancement would make the role and integration of FPCC within our overall pipeline more apparent.
>
> **Regarding the selection of $P_{p}^{\(1\)}$ and $P_{p}^{\(2\)}$:**
>
> Similar to prior self-supervised approaches, our method trains a separate network for each category, for example, chair or plane. Thus, $P_{p}^{\(1\)}$ and $P_{p}^{\(2\)}$ are always randomly selected from different samples within the same category, rather than across categories. Moreover, combining samples from the same category is more reasonable, for instance, combining a plane and a chair would yield unnatural results. Consequently, there is no risk of introducing cross-category bias, and our experimental design ensures that the FPCC is applied within a consistent semantic context.
>
> **Q3:** What network architectures constitute the two disentanglers in the Feature Disentanglement Module? How can we interpret that one models shape information while the other models detail information?
>
> **Network Architectures of the Disentanglers:**
>
> Both disentanglers are implemented as Multi-Layer Perceptrons (MLPs) with a single hidden layer, following the implementation in OptDE (as cited in [11] of our manuscript).
>
> **Interpretation of Shape and Detail (Style) Modeling:**
>
> In our framework, “shape” refers to the global structure of a point cloud, which is captured by the skeleton points obtained through farthest point sampling. In contrast, “detail” (or “style”) refers to the local geometric features, represented by the K-nearest neighbor points around each skeleton point. Based on this setup, we generate $P_{recombined}$ from $P_p^{(1)}$ and $P_p^{(2)}$, as shown in Figure 3. As we mentioned in **Q2** , the $P_{recombined}$ has the same shape as $P_p^{(1)}$, so we use the shape disentangler to extract shape features from both and ensure shape feature consistency. Similarly,  $P_{recombined}$ has the same style as  $P_p^{(2)}$, so we use the style disentangler to extract style features from both and ensure style feature consistency. These two consistency constraint encourage each disentangler to focus on its designated attribute: the shape disentangler models the global structure, while the style disentangler models local details.
>
> To further enhance clarity, we will update Figure 3 in our revised manuscript to explicitly include both disentanglers. Additionally, we will revise the caption of Figure 3 to more clearly articulate the purpose of FPCC, emphasizing its role in enforcing effective feature disentanglement.
>
> **Q4:** Which component in the Pipeline corresponds to the "lagg" section mentioned in line 173? It is recommended that the PPM module be presented in a separate, more detailed figure.
>
> In our current implementation, we have combined the losses $ \cal L_{agg}$ and $ \cal L_{info}$ into a unified loss term $ \cal L_{cluster}$, as shown in Eq.(6), which is explicitly depicted in the Pipeline (Figure 2). As a result, the $ \cal L_{agg}$ is not independently illustrated, which may have affected the clarity of our presentation. We fully agree with your suggestion to present the PPM module in a separate, dedicated figure. In the revised manuscript,  we will simplify the component of the PPM in Figure 2 and introduce a new, standalone figure for the PPM, similar to Figure 3. In this new figure, we will detail the computation pipelines for both loss terms ($ \cal L_{agg}$ and $ \cal L_{info}$), thereby providing a clearer and more comprehensive illustration of the PPM’s structure and its associated losses.
>
> **Q5:** In Section 3.3, labeling the approach employing CD loss as "Instance-level Contrastive Learning" may not be rigorously accurate. Adopting an alternative term would not undermine the novelty of this work.
>
> Thank you for your valuable feedback and for recognizing the novelty of our approach. Our initial use of the term **“Instance-level Contrastive Learning”** was motivated by the design of the $\cal L_{instance}$ : this loss computes the distance between two completion results generated from different partial observations of the same instance. By minimizing this discrepancy, we encourage the network to treat these two outputs as positive pairs, promoting consistency and the preservation of instance-specific details.
>
> Nevertheless, we agree that the use of the term “contrastive learning” may not be rigorously precise in this context, as our approach does not involve negative pairs or the standard contrastive learning framework. To improve clarity without affecting the novelty of our work, we  would  adopt the alternative term such as **“instance-consistency loss”** .
>
> **Q6:** The abbreviated loss terms (Lcom, Lins) are used throughout the paper, while Figure 2 employs full terms. This inconsistency is inappropriate. Consistency should be maintained.
>
> Thanks. We will revise the manuscript so that the loss terms are presented consistently throughout. Specifically, we will use $\cal L_{cluster}$ and $\cal L_{instance}$ both in the main text and in all figures.

---

> > ### Comment · Reviewer_KtxY · 2025-08-05
> > **-**
> >
> > Most of my concerns have been addressed. The authors could consider incorporating the suggested improvements to the figures in the final version.

---

> ### Author Response · Authors · 2025-08-05
>
> Thank you for your valuable comments. We are glad that your concerns have been addressed and will make sure to further enhance the figures as suggested in the final submission.

---

### Decision · Program_Chairs · 2025-09-17

**Decision:**

Accept (spotlight)

**Comment:**

This paper presents a self-supervised framework for point cloud completion that explicitly reconstructs complete structures from incomplete data by employing dual-level contrastive learning to disentangle shape and style features. Reviewers find the approach novel and empirically strong across synthetic and real-world benchmarks, showing clear improvements over prior self-supervised methods. Main concerns of this work lie in presentation clarity, missing ablations and details, and limited discussion of related zero-shot or diffusion-based methods. Since these issues are well addressed and clarified during rebuttal, the contribution of this work is considered meaningful and the paper is clearly above the acceptance threshold.